# External Scaffold for Venous Graft to Treat Chronic Limb-Threatening Ischemia: Results of the FRAME Vascular Support

**DOI:** 10.3390/jcm13072095

**Published:** 2024-04-03

**Authors:** Nunzio Montelione, Vincenzo Catanese, Antonio Nenna, Teresa Gabellini, Chiara Ferrisi, Julia Paolini, Alessandro Ciolli, David Barillà, Francesco Loreni, Massimo Chello, Francesco Spinelli, Francesco Stilo

**Affiliations:** 1Vascular Surgery, Fondazione Policlinico Universitario Campus Bio-Medico, 00128 Rome, Italy; n.montelione@policlinicocampus.it (N.M.); v.catanese@policlinicocampus.it (V.C.); teresa.gabellini91@gmail.com (T.G.); j.paolini@policlinicocampus.it (J.P.); a.ciolli@policlinicocampus.it (A.C.); f.spinelli@policlinicocampus.it (F.S.); f.stilo@policlinicocampus.it (F.S.); 2Cardiac Surgery, Fondazione Policlinico Universitario Campus Bio-Medico, 00128 Rome, Italy; chiara.ferrisi@unicampus.it (C.F.); francesco.loreni@unicampus.it (F.L.);; 3Vascular Surgery Residency Program, University of Ferrara, 44124 Ferrara, Italy; 4Cardiac Surgery Residency Program, Università Campus Bio-Medico di Roma, 00128 Rome, Italy; 5Vascular Surgery Residency Program, Università Cattolica del Sacro Cuore, 00128 Rome, Italy; 6Vascular Surgery, Humanitas Clinical and Research Center—IRCCS, Rozzano, 20148 Milan, Italy

**Keywords:** chronic limb-threatening ischemia, scaffold, external, support, FRAME, vein, graft

## Abstract

**Background**: The surgical treatment of chronic limb-threatening ischemia and optimal conduit choice are extensively debated. The presence of suboptimal autologous material, such as varicosities or venous aneurysms, might impair long-term outcomes. Therefore, kink-resistant external supports have been advocated in the recent literature to improve the conduit quality and outcomes. This study analyzes the FRAME external support in venous ectasic grafts in patients with chronic limb-threatening ischemia. **Methods**: From September 2017 to September 2023, a total of sixteen patients underwent CLTI surgery with FRAME external support for venous grafts. The inclusion criteria for FRAME applications were varicose or ectasic autologous material with a diameter ≥ 4.5 and ≤ 8 mm in an isolated segment or in the entire vein and a higher risk of bypass extrinsic compression (e.g., extra-anatomical venous bypass course). **Results**: Technical success and intraoperative patency were achieved in all cases. At 30 days, the limb salvage and survival rates were 100%. The primary bypass patency was 93.7% due to an early graft occlusion. No graft infection was registered. In one case, dehiscence of the surgical wound was treated by surgical debridement and antibiotic therapy. Minor amputation was required in four patients. Over a median follow-up of 32 months, two occlusions were observed; one was treated with reoperation and the other with major amputation. The primary patency was 68.7% and the assisted primary patency was 75%. Limb salvage rates observed during the entire follow-up period were 87.5%. No graft infections or dilatation of the reinforced veins were registered. **Conclusions**: For patients with CLTI undergoing infrainguinal bypass, satisfactory results in terms of patency and limb salvage rates were achieved using the autologous venous material, even if ectasic or varicose, with the vascular external support FRAME.

## 1. Introduction

Chronic limb-threatening ischemia (CLTI) is associated with mortality, amputation, and impaired quality of life, and its optimal treatment remains debated [1,2,3,4,5,6]. As recently reported in the Global Vascular Guidelines, vein bypass should be preferred for patients with advanced limb threat and high-complexity disease in cases of low–intermediate surgical risk [1]; however, the optimal revascularization strategy is also influenced by the availability of autogenous vein for open bypass surgery [1]. Several studies highlight that, for below-the-knee revascularization, the great saphenous vein (GSV) should be the first-choice conduit, thanks to its greater long-term patency and lower infectious risk compared to the use of prosthesis bypass (PTFE) [7,8,9,10]. However, according to the literature, in 20–40% of the cases, the GSV is unavailable because it has been previously excised or is unsuitable [11,12].

When GSV is unsuitable, other veins, such as the short saphenous vein (SSV) and arm veins, should be harvested for bypass grafts, leading to satisfactory short- and long-term patency rates [13,14,15]. The same results also apply when using an autogenous composite vein [16]. However, it may occur that the autologous material is suboptimal because of the presence of varicosities and ectasias that cause progressive thickening of the venous graft wall. Hence, the bypass is more exposed to aneurysm formation and graft failure [17,18,19].

The vein quality concept was first reported in late 70s and gradually recognized as a determinant of vein bypass failure [20]. The authors stated that the quality of the autologous vein was one of the most crucial aspects affecting the long-term patency of the bypass graft. However, no precise definition of vein quality has been formulated, although size, morphology, and compliance are all relevant criteria. In general, lower limb bypass requires longer grafts than coronary artery bypass grafting, which may result in a higher proportion of GSVs being unsuitable for grafts.

In cases of ectasic or varicose veins, associated chronic venous hypertension initiates a range of pathophysiologic changes in the venous wall and surrounding tissues, including inflammation and increased permeability of the endothelium. The resulting accumulation of fibrin and hemosiderin in the perivascular tissues acts to exacerbate inflammation and promote collagen synthesis by fibroblasts, leading to venous wall thickening and remodeling [21,22]. Tissue hypoxia leads to apoptosis and extracellular changes [23]. Aging-induced inflammation in veins is associated with elevated cytokine production and increased MMP activation, which likely play a critical role in the pathogenesis of chronic venous disease [22,23].

The aberrant wall structure of varicose veins and impaired functional capabilities of endothelial cells can lead to hemodynamic disturbance, increasing the long-term risk of thrombosis, and the development of aneurysms, which can burst when exposed to arterial pressure [24,25].

In some studies, it has been shown that the arm veins used in lower limb bypass procedures could lead to stenosis and aneurysms [26,27]. This scenario is usually considered a contraindication for autologous vein grafting [28].

To find solutions allowing the use of autologous material despite its low quality, numerous studies on animals have been conducted to evaluate the effectiveness of external stents of various materials and structures positioned around the venous graft. Initial results of those experimental studies showed how the external devices act by containing the graft, avoiding postoperative dilation, preventing turbulent blood flow, and improving long-term patency. A stimulus to angiogenesis and the formation of vasa vasorum through the accumulation of fibrinous exudate between the sheath and the adventitia of the vessel was also observed, which significantly reduces the intimal hyperplasia [21,22,23,24].

Following this evidence, studies were also conducted on humans, involving the use of a polyester mesh with acceptable short-term patency [29,30,31,32,33]. Also, the use of venous bypass procedures that circumvent the “normal” anatomic pathways could expose the graft to failure due to possible compression.

The aim of our study is to evaluate the effectiveness of a kink-resistant external support (FRAME ^TM^, Vascular Graft Solutions Ltd., Tel Aviv, Israel) in venous bypasses performed with varicose or ectasic autologous material in patients with limb ischemia and its value in cases of extra-anatomical vein graft positioning.

## 2. Patients and Methods

### 2.1. Study Design

During the study period between September 2017 and September 2023, patients with venous bypasses performed using the external vascular support FRAME ^TM^ as scaffolding were enrolled. The indications for surgery were (i) CLTI defined according to the Global Vascular Guidelines [1] referring to the Wound, Ischemia, and foot Infection (WIfI) classification and (ii) arm ischemia.

### 2.2. Inclusion Criteria

The inclusion criteria for FRAME ^TM^ applications were patients with dilated veins or with focal ectasias or patients with vein abnormalities at surgical inspection. In all patients, the quality of vein material was suboptimal due to varicosities or ectasic dilatation [29]. To be included in the study, patients should have all of the following:(1)vein diameter ≥4.5 and ≤8 mm(2)diffusely varicose vein/severe ectasia in an isolated segment (with reduced thickness)(3)higher risk of bypass extrinsic compression (e.g., extra-anatomical course).

The exclusion criteria were grossly infected tissues in the affected limb and reduced life expectancy (<1 year) due to oncologic, cardiologic or neurologic disorders that would have affected long-term results.

The study received approval from the Institutional Review Board of Fondazione Policlinico Universitario Campus Bio-Medico di Roma (code UCBM/20231002/NM1, approved on 10 February 2023). All patients gave their written consent to the procedure.

The demographic parameters and risk factors of the enrolled patients are listed in Table 1.

All patients underwent a preoperative specialist visit, which included the patient’s clinical history, the objective examination focusing on the arterial wrists and the presence/absence of lesions, and ultrasound arterial mapping. With the ultrasound, veins were assessed by measuring the diameters of the GSVs and short saphenous veins (SSVs) throughout the limb in the orthostatic position. Also, the veins of the upper extremities (cephalic and basilic veins) were evaluated if necessary.

Moreover, the veins of the arms were studied using proximal compression. Selective preoperative computed tomography angiography (CTA) or invasive catheter angiography was also performed.

### 2.3. Material Description

The material used for the study is the new braided cobalt-chrome kink-resistant external support (FRAME ^TM^, Vascular Graft Solutions Ltd., Tel Aviv, Israel). The device is very thin (50 μm) and flexible. FRAME ^TM^ does not require glues or sutures for its application and its use reduces the incidence of occlusive thrombosis and neointimal hyperplasia (Appendix A). In addition, this device prevents the irregular post-dilatation of the vein graft and guarantees a constant flow pattern throughout the vein lumen.

### 2.4. Description of Variables

FRAME ^TM^ is available in different diameters (A = 4.2 mm, B = 4.6 mm, C = 5.2 mm, D = 5.6 mm) and lengths (from 10 cm to 90 cm), and it is also possible to arrange the final length “in situ” by cutting it before the distal anastomosis is performed.

### 2.5. Surgical Preparation

In cases of localized dilatations of a segment of the GSV, a partial bypass scaffolding (i.e., 10 to 15 cm) was used, to keep a healthy venous tract free from scaffolding and avoid additional and unnecessary maneuvers. Indeed, patients with a long segment of varicose vein underwent a completely covered ex situ vein bypass (Figure 1A). During the surgical preparation, a single surgical incision is made along the entire course of the vein, which is harvested and devalvulated. With this technique, the surgeon can control the course of the vein, close any arteriovenous fistulas, avoid twisting and movement of the vein and the FRAME ^TM^ used. Anatomical tunneling has been avoided.

Once the target vein was harvested, gradual dilation was achieved to evaluate any damage on the lateral branches and the quality of the autologous material. Subsequently, bench vein devalvulation was accomplished using a Chevalier valvulotome. The vein was then marked every 10 cm to calculate its length and to avoid twisting of the graft during the external support positioning. The size of the external mesh (FRAME ^TM^) was based on the diameter measurement of the dilated graft using a dedicated “surgical ruler” (Figure 1B,C).

At this stage, the FRAME ^TM^ was positioned by sliding it along the vein with the appropriate cannula (Figure 1D) (Appendix A). When the FRAME ^TM^ expands, the mesh is adjusted for the entire length of the vein. Finally, once the proximal anastomosis was carried out and the exact length of the venous graft and FRAME ^TM^ (Figure 2A) was assessed in situ, the distal anastomosis was performed (Figure 2B).

### 2.6. Outcomes and Follow-Up

The main outcome of the surgery is to evaluate the technical success (namely the effectiveness of the procedure), analyzing the correct packing of the bypass and the intraoperative bypass patency. At 30 days after the surgery and during the follow-up, the primary and primary assisted bypass patency rates were evaluated, as well as the limb salvage rate.

The primary bypass patency was defined as the time (in months) from the initial restoration of vessel patency (index procedure) to any secondary intervention, preserving bypass patency. The primary-assisted patency of the bypass was defined as the time (in months) from the initial procedure to any failure of the bypass, which required additional treatment due to a significant stenosis, but not in the case of full thrombosis or occlusion of the graft. External support infection rates, upon reoperation for surgical wound infection, as well as graft aneurysmal dilatation over time, were also analyzed.

Follow-up was performed at 30 days, 3 months, 6 months, and every 6 months thereafter by clinical examination and morphological evaluation of the bypass using color Doppler ultrasound. In selected cases, postoperative CTA was performed.

### 2.7. Statistical Analysis

In the text and tables, data are shown as number (percentage) or median (interquartile range). Considering the small number of patients, no dedicated time-to-event analysis was performed and only crude outcomes are shown.

## 3. Results

During this study period, 16 patients underwent vein graft bypass using the external vascular support FRAME ^TM^ as scaffolding. For 13 patients (81.2% of the sample), the use of FRAME ^TM^ was driven by the presence of ectasic or varicose veins; in 3 patients (18.8%), the external vascular support was used with the aim to avoid compression, considering the extra-anatomical course of the venous graft: (i) in one patient with arm ischemia, an axillary-brachial extra-anatomical vein graft bypass was performed due to skin infection after shoulder prosthesis [30]; (ii) in one patient, an external iliac artery to deep femoral artery venous graft was placed in an extraanatomical way, avoiding the infected field in the groin due to a previous common femoral to deep femoral artery ePTFE bypass; (iii) for the third patient a superficial femoral artery to anterior tibial artery venous graft was tunneled in a lateral position, avoiding an infected wound in the medial region of the leg (Figure 3A,B).

Out of the 13 patients treated for CLTI, 8 (61.5%) were at WIfI clinical stage 3, while 5 patients (38.5%) were at clinical stage 4. In the whole study group, general anesthesia was performed in 5 patients (31.2%), and in the remaining 11 patients (68.8%), spinal anesthesia and/or peripheral nerve blockage were performed. Mean diameter of autologous vein material was 6.8 mm (range 3.8–8 mm), with FRAME ^TM^ C and D used in 3 (23%) and 13 (77%) patients, respectively.

In 13 patients (81.2%), the autologous bypass was performed with the great saphenous vein, in one (6.2%) case with the short saphenous vein (SSV), in one case with a composition of both, and in one case using arm vein. In CLTI patients, 9 bypasses (69.2%) were performed below the knee and 3 (23.1%) below the ankle; intraoperative details are reported in Table 2. Technical success was achieved in all cases and the mean surgical time was 360 (280–425) min. The mean length of stay was 9 (7–11) days.

At 30 days, the limb salvage and survival rates were 100%. The primary bypass patency was 93.7% due to an early graft occlusion. No graft infection was registered. In one case, dehiscence of the surgical wound was treated by surgical debridement and antibiotic therapy. Minor amputation was required in 4 patients (25%).

During a median follow-up of 32 (26–38) months (range 3–72), two patients died (one from acute coronary syndrome 2 months after the index procedure, and one from lung cancer after 2 years). Two occlusions were registered at 6 and 13 months. In one patient, reintervention was necessary 4 months from the index procedure due to a focal stenosis in the proximal tract of the bypass, treated with an enlargement patch using the cephalic vein and restoration of normal flow (Figure 4). In one patient, major amputation was required secondary to graft occlusion. The primary patency was 68.7% and the assisted primary patency was 75%. Limb salvage rates observed during the entire follow-up period were 87.5%. No graft infections or dilatation of the arterialized veins were registered. A CT angiography was performed in selected patients 2 weeks after the procedure (Figure 5).

## 4. Discussion

Infrainguinal bypass is the first-choice treatment in patients with CTLI and extensive atherosclerotic lesions with an available GSV and acceptable surgical risk [1,34]. Several studies report satisfactory results with alternative autologous materials, such as the SSV or arm veins, thanks to their higher long-term patency and lower infectious risk compared to the use of prosthetic material [8,13,14,15].

However, when autologous venous material is varicose, the bypass could be more susceptible to the formation of aneurysms or dilatation and subsequent failure. Still today, there is little evidence regarding bypasses performed with varicose venous material and external vascular support. In addition, there are very few studies that have assessed the value of FRAME ^TM^ as a new scaffolding material in peripheral vascular bypass [30,32] as analyzed in our series.

To find solutions allowing the use of autologous material despite its low quality, numerous studies on animals have been conducted to evaluate the effectiveness of external stents of various materials and structures positioned around the venous graft. As reported [21], the use of stents and external synthetic sheaths in porcine vein models used as grafts has a remarkable effect on the remodeling and thickening of the venous graft. This research showed how these external devices act by containing the graft, avoiding postoperative dilation, preventing turbulent blood flow, and improving long-term patency [21,22]. Furthermore, it should be noted that the presence of the stent stimulates angiogenesis and the formation of neo vasa vasorum through the accumulation of fibrinous exudate between the sheath and the adventitia of the vessel, significantly reducing intimal hyperplasia. The benefit is amplified by the features of the device: the presence of a wide-meshed porous stent facilitates the penetration of the vasa vasorum to the venous wall. The external stent must be positioned loosely to allow normal expansion of the vein due to blood pressure, as a restrictive stent could cause the opposite effect by increasing the intimal hyperplasia of the graft [7]. Further studies also highlighted that the external stent reduces, in addition to the formation of the neointima, the appearance of paravascular nerve bundles originating from the adventitia, which had a role in venous graft failure [23,24].

Once the success of these techniques was demonstrated, they were also used on humans using different materials. A multicenter study by Arvela et al. highlighted how the polyester mesh (PTFE) is safe and feasible in addition to peripheral revascularization, allowing the use of suboptimal venous grafts with acceptable short-term patency [25]. Experimental studies have shown that a considerable reduction in the diameter of veins can be achieved by external wrapping without the development of obstructing folds of the vein wall and with satisfactory patency rates [26,27,28].

In the large series published by Carella et al. [29] in 2011, a multifilament polyester mesh (polyethylene terephthalate: ProVena; BBraun Aesculap, Tuttlingen, Germany) was used for the external scaffolding of varicose or ectasic autologous veins. Among the 21 patients enrolled in the study, the primary, assisted patency, and amputation-free survival rates at 24 months were 57.1% (SE ± 3.9), 81% (SE ± 3.2), and 85.7% (SE ± 2.8), respectively. The authors reported that in the other bypasses performed in the same period and without mesh, the primary patency was 63.8%, secondary patency was 80.5%, and the amputation-free survival rate was 89.3% at 24 months, without statistically significant differences between the groups.

According to already published studies [33], external stenting with macro-porous polyester mesh reduced neointimal hyperplasia more effectively than PTFE and other commercially available stents, possibly due to better circumferential compliance. Although it has been demonstrated in several animal studies that external support reduces intimal hyperplasia, the ideal scaffolding material has not yet been identified.

In recent years, new materials were produced and marketed as external supports for varicose veins, such as the kinking-resistant cobalt-chrome outer mesh (FRAME ^TM^, Vascular Graft Solutions Ltd., Israel); despite this, there is currently no specific indication in the guidelines based on the available literature.

In our study, 13 patients enrolled had a diagnosis of CLTI with resting pain and/or tissue loss. Among these, 12 patients underwent bypass surgery with an autologous graft coated with FRAME ^TM^ due to the poor quality of the venous material, which appeared ectasic and varicose, also using the SSV in two cases and the arm vein in one case with satisfactory patency and limb salvage rates at follow-up. The vein graft material used in the present series was non-optimal, and therefore our results can be considered acceptable compared to results obtained from previous experiences [25]. Indeed, the primary patency at 32 months of follow-up was 68.7%, but the assisted primary patency was 75%, which emphasizes the importance of vein graft surveillance. Duplex scan surveillance seems to be especially important in cases of suboptimal veins used and arm vein grafts. In infrainguinal vein bypasses, the incidence of focal stenoses due to intimal hyperplasia is reported to be about 20–35% within 1–2 years after operation [34,35] and the incidence should be higher in cases of non-optimal autologous vein.

Also, it is interesting to note that secondary open procedures, as reported in one case treated with venous patch angioplasty because of a focal tight stenosis, appear possible.

According to previous studies [29], in our experience the use of external scaffolding made the procedure more complex and time-consuming, with a mean surgical time of 360 min, also due to the challenges in harvesting the varicose veins. However, this new vascular support appears easy to apply and to use in different kinds of surgical fields.

Indeed, during the use of ProVena, disruption of the valves of totally covered non-reversed vein grafts is more difficult as valve pockets are no longer visible using this polyester mesh. Although some authors [25] overcame this problem by initially covering only a short segment of the proximal graft, while the rest is covered after completion of the proximal anastomosis and valve lysis, this process should be challenging and time consuming. On the contrary, using the new vascular support FRAME ^TM^, after on-the-bench valvulotomy, the external scaffolding could be applied directly along the ex situ, inflated vein.

Another important aspect is the potential role of avoiding compression in extra-anatomical venous grafts. Starting from an already reported experience [30], in which FRAME was used to scaffold a GSV axillary-brachial bypass in an extra-anatomical fashion to avoid scar tissue and infected fields in the shoulder, we decided to use this technique in two other cases with satisfactory results.

In the present series, despite the use of FRAME ^TM^ in CLTI patients with extensive gangrene or cases of documented infected fields, there were no infections related to the external vascular support or to the vein graft. This kind of material (cobalt-chrome), should probably be better in these kinds of patients than the polyester mesh or PTFE that can be prone to infections, as reported in some experiences, especially when used in CLI patients with extensive gangrene [11].

However, the polyester external mesh did not impose asymmetry on the graft due to the absence of relative rigidity that theoretically could have happened with the stent structure of FRAME ^TM^, even if this event was not reported in the present experience.

Molecular therapies such as mesenchymal stem cells or gene therapy are currently being investigated in this field and will hopefully improve outcomes [36,37,38,39], but until the widespread use of those innovative techniques in clinical practice, it remains advisable to use team-based care to provide the best treatment option in each patient to deal with any anatomic difficulties such as ectasic veins. Close collaboration among surgeons, clinicians and biomedical engineers is crucial to tailor the treatment for each patient [40,41], similarly to the lessons of the Heart Team in cardiology. In this scenario, patient-reported outcomes play a pivotal role in evaluating the cost-effectiveness of each treatment [2,5,42] as well as circulating biomarkers to promptly detect complications even before clinical events [43].

## 5. Conclusions

In our experience, for CLTI patients undergoing infrainguinal bypass, satisfactory results in terms of patency and limb salvage rates were achieved using the autologous venous material, even if ectasic or varicose, with the new vascular external support FRAME ^TM^. The use of extra-anatomical venous bypass, combined with a kink-resistant external vascular support, seems to be a feasible solution for the treatment of patients showing upper or lower limb ischemia and without a viable anatomical pathway.

## Figures and Tables

**Figure 1 jcm-13-02095-f001:**
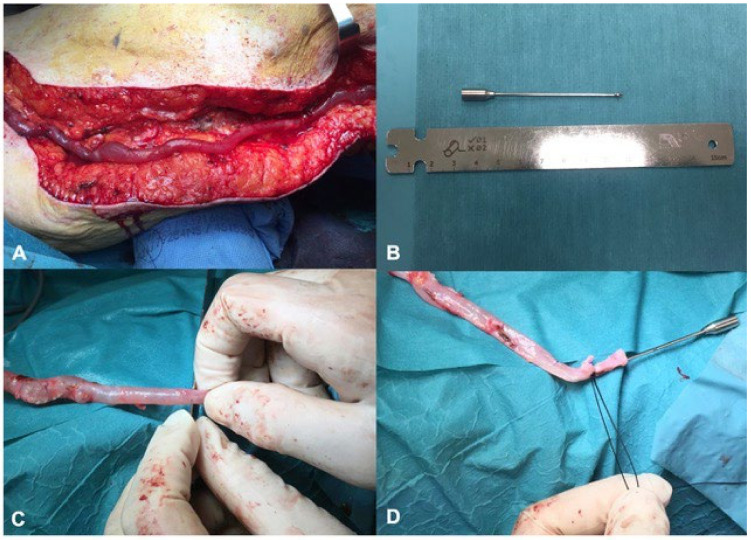
Long segment of varicose vein (**A**); surgical ruler and olive cannula for vein measurement and FRAME application (**B**); vein measurement (**C**) and olive cannula fixation at the distal end of the vein graft for FRAME application (**D**).

**Figure 2 jcm-13-02095-f002:**
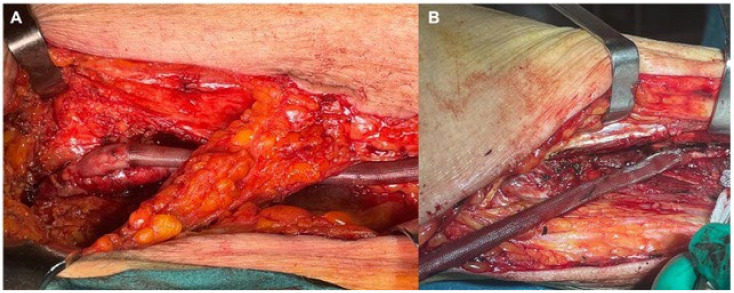
Intraoperative details showing proximal and distal anastomoses at the common femoral (**A**) and peroneal artery (**B**) anastomosis in one of the chronic limb-threatening ischemia patients using FRAME ^TM^ as scaffolding.

**Figure 3 jcm-13-02095-f003:**
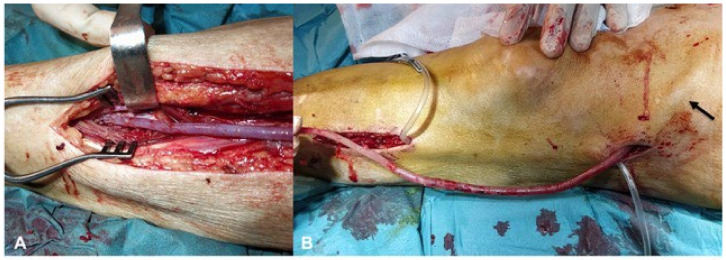
Distal anastomosis in the anterior tibial artery (**A**) and the extra-anatomical course (**B**) of the venous graft that crosses the subcutaneous tissue (arrow) in the middle portion of the thigh.

**Figure 4 jcm-13-02095-f004:**
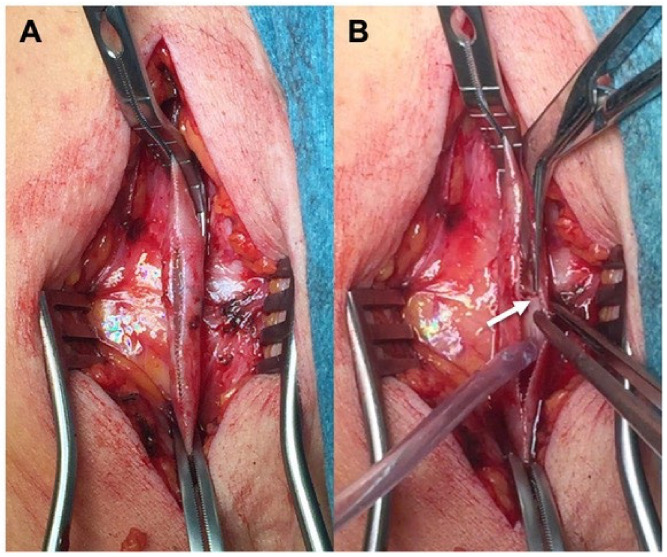
Reintervention in a femoro-posterior tibial Great Saphenous Vein graft in which the external scaffolding was easily clamped and opened (**A**) for the treatment of a focal restenosis (arrow) during follow-up (**B**).

**Figure 5 jcm-13-02095-f005:**
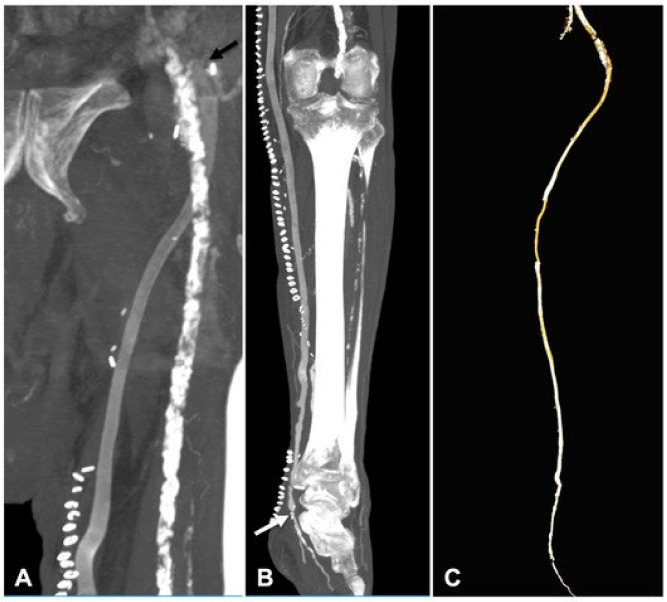
CT angiography performed 2 weeks after surgery that showed the proximal anastomosis in the common femoral artery (**A**) and the below-the-ankle anastomosis in the posterior tibial artery (**B**) in a case with FRAME; (**C**) three-dimensional reconstruction underlining the external scaffolding material, bypass patency and the satisfactory distal perfusion to the foot.

**Table 1 jcm-13-02095-t001:** Demographic data and baseline characteristics.

Age, Median (Range)	74.5	(65–91)
Male (n, %)	13	81%
Smoking (n, %)	11	69%
Hypertension (n, %)	12	75%
Diabetes (n, %)	7	43%
Hyperlipidemia (n, %)	8	50%
Coronary Artery Disease (n, %)	3	19%
Chronic Kidney Disease stage III + (n, %)	1	6%

**Table 2 jcm-13-02095-t002:** Operative details of bypass surgery with FRAME external support.

Indication for Surgery	Anesthesia	Inflow Artery	Outflow Artery	Vein Graft Details
CLTI	GA	CFA	POP	LSV
CLTI	SA	CFA	PER	GSV + LSV
CLTI	SA	CFA	PTA	GSV
CLTI	GA	DFA	PTA	GSV
CLTI	SA + LRA	CFA	PTA	GSV
CLTI	SA	CFA	ATA	GSV
CLTI	SA + LRA	CFA	ATA	GSV
CLTI	SA + LRA	SFA	PTA	GSV
CLTI and Infection	GA	EIA	DFA	GSV
ALI	GA	Axillary A	Brachial A	GSV
CLTI	SA + LRA	SFA	PTA	GSV
CLTI	GA	CFA	ATA	Cephalic V
CLTI	SA + LRA	CFA	PTA	GSV
CLTI	SA + LRA	SFA	PER	GSV
CLTI	SA + LRA	CFA	PTA	GSV
CLTI	SA + LRA	CFA	ATA	GSV

GA = general anesthesia; SA = spinal anesthesia; LRA = loco-regional Anesthesia; CFA = common femoral artery; DFA = deep femoral artery; SFA = superficial femoral artery; POP = popliteal artery; PER = peroneal artery; PTA = posterior tibial artery; ATA = anterior tibial artery; LSV = lesser saphenous vein; GSV = great saphenous vein.

## Data Availability

Data are contained within the article and Appendix A. All details are available upon request to the corresponding author.

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
