# Peer review of "External Scaffold for Venous Graft to Treat Chronic Limb-Threatening Ischemia: Results of the FRAME Vascular Support"

_jcm, 2024, doi:10.3390/jcm13072095_

Round 1
Reviewer 1 Report
Comments and Suggestions for Authors
Congratulations for your interesting work regarding a case series of 16 patients with external scaffold of vein grafts used for bypass. Your manuscript is comprehensive and well-written. Some suggestions for improvent: 1) Please mention if any consent was obtained from the patients, 2) Why did you assume GSVs with a diameter more than 4.5mm as pathological/needing scaffolding, while GSVs could be considered as normal up to a 6mm diameter?, 3) Please separate your methods part in smaller parts with subtitles (eg. study design, inclusion/exclusion criteria, material description, description of variables, statistical analysis, etc.)
Author Response
Reviewer 1
Congratulations for your interesting work regarding a case series of 16 patients with external scaffold of vein grafts used for bypass. Your manuscript is comprehensive and well-written. Some suggestions for improvement.
Thank you for your comments and your appreciation.
Please mention if any consent was obtained from the patients.:
Thank you. Written informed consent was given by all patients. This statement has been addend in the revised manuscript.
Why did you assume GSVs with a diameter more than 4.5mm as pathological/needing scaffolding, while GSVs could be considered as normal up to a 6mm diameter?
Thank you for your suggestion. The statement has been changed avoiding misleading. We agree with the reviewer that GSVs can be normal up tp 6 mm in diameter, but focal ectasia and varicosities play a major role in wall weakness. We remodulated the statement, and inclusion criteria are similar to already published clinical trials (i.e. vein diameter ≥ 4.5 and ≤ 8mm + diffusely varicose vein/severe ectasia in isolated segment with reduced thickness + higher risk of bypass extrinsic compression such as extra-anatomical course).
Please separate your methods part in smaller parts with subtitles (eg. study design, inclusion/exclusion criteria, material description, description of variables, statistical analysis, etc.).
According to your valuable suggestion, methods has been separated in smaller parts with subtitles.
Reviewer 2 Report
Comments and Suggestions for Authors
**L52-53: The authors commented that varicosity causes progressive graft thickening. Please provide the data to support, such as patency of bypass graft with veins with varicosities as the most of the cases used FRAME in this series was due to ectasia of the vein. This study does not have comparison control group and literature support should be presented.
L98-105: Please provide the thickness of the FRAME system. Why not use the 90cm to all the cases if the external support system provides better outcome?
L108-109: When using the short segment FRAME, how can surgeons ensure this FRAME does not move when pulling through the graft in the tunnel? Do authors not use anatomical tunneling?
Author Response
Reviewer 2
L52-53: The authors commented that varicosity causes progressive graft thickening.
Please provide the data to support, such as patency of bypass graft with veins with varicosities as the most of the cases used FRAME in this series was due to ectasia of the vein. This study does not have comparison control group and literature support should be presented.
Thanks for your valuable suggestions. Literature support has been added in the revised manuscript.
L98-105: Please provide the thickness of the FRAME system.
The thickness of the FRAME has been added in the revised version as per your suggestion.
Why not use the 90cm to all the cases if the external support system provides better outcome?
Thank you for your question. As reported in text, in case of fully ectasic vein, a whole coverage of the vein graft has been provided. In case of isolated varicose segment, the longer FRAME system has been cut according to the length of the “diseased” vein graft. This is because the Authors prefer to keep a healthy venous tract free from scaffolding avoiding additional and unnecessary maneuvers on the graft.
L108-109: When using the short segment FRAME, how can surgeons ensure this FRAME does not move when pulling through the graft in the tunnel? Do authors not use anatomical tunneling?
Thank you. A detailed description of the technique in now added in the revised manuscript, please find it in the Surgical preparation section. Anatomical tunnelling has never been used.